# Rational neural networks

**Nicolas Boullé**
Mathematical Institute
University of Oxford
Oxford, OX2 6GG, UK
boulle@maths.ox.ac.uk

**Yuji Nakatsukasa**
Mathematical Institute
University of Oxford
Oxford, OX2 6GG, UK
nakatsukasa@maths.ox.ac.uk

**Alex Townsend**
Department of Mathematics
Cornell University
Ithaca, NY 14853, USA
townsend@cornell.edu

## Abstract

We consider neural networks with rational activation functions. The choice of the nonlinear activation function in deep learning architectures is crucial and heavily impacts the performance of a neural network. We establish optimal bounds in terms of network complexity and prove that rational neural networks approximate smooth functions more efficiently than ReLU networks with exponentially smaller depth. The flexibility and smoothness of rational activation functions make them an attractive alternative to ReLU, as we demonstrate with numerical experiments.

## 1   Introduction

Deep learning has become an important topic across many domains of science due to its recent success in image recognition, speech recognition, and drug discovery [23, 28, 29, 32]. Deep learning techniques are based on neural networks, which contain a certain number of layers to perform several mathematical transformations on the input. A nonlinear transformation of the input determines the output of each layer in the neural network: $x \mapsto \sigma(Wx + b)$, where $W$ is a matrix called the weight matrix, $b$ is a bias vector, and $\sigma$ is a nonlinear function called the activation function (also called activation unit). The computational cost of training a neural network depends on the total number of nodes (size) and the number of layers (depth). A key question in designing deep learning architectures is the choice of the activation function to reduce the number of trainable parameters of the network while keeping the same approximation power [17].

While smooth activation functions such as sigmoid, logistic, or hyperbolic tangent are widely used, they suffer from the "vanishing gradient problem" [5] because their derivatives are zero for large inputs. Neural networks based on polynomial activation functions are an alternative [9, 11, 19, 20, 33, 52], but can be numerically unstable due to large gradients for large inputs [5]. Moreover, polynomials do not approximate non-smooth functions efficiently [51], which can lead to optimization issues in classification problems. A popular choice of activation function is the Rectified Linear Unit (ReLU) defined as $\text{ReLU}(x) = \max(x, 0)$ [26, 39]. It has numerous advantages, such as being fast to evaluate and zero for many inputs [16]. Many theoretical studies characterize and understand the expressivity of shallow and deep ReLU neural networks from the perspective of approximation theory [13, 31, 36, 49, 53].

ReLU networks also suffer from drawbacks, which are most evident during training. The main disadvantage is that the gradient of ReLU is zero for negative real numbers. Therefore, its derivative is zero if the activation function is saturated [34]. To tackle these issues, several adaptations to ReLU have been proposed such as Leaky ReLU [34], Exponential Linear Unit (ELU) [10], Parametric Linear Unit (PReLU) [22], and Scaled Exponential Linear Unit (SELU) [27]. These modifications outperform ReLU in image classification applications, and some of these activation functions have trainable parameters, which are learned by gradient descent at the same time as the weights and biases of the network. To obtain significant benefits for image classification and partial differential equation (PDE) solvers, one can perform an exhaustive search over trainable activation functions constructed

from standard units [25, 48]. However, most of the "exotic" activation functions in the literature are motivated by empirical results and are not supported by theoretical statements on their potentially improved approximation power over ReLU.

In this work, we study rational neural networks, which are neural networks with activation functions that are trainable rational functions. In Section 3, we provide theoretical statements quantifying the advantages of rational neural networks over ReLU networks. In particular, we remark that a composition of low-degree rational functions has a good approximation power but a relatively small number of trainable parameters. Therefore, we show that rational neural networks require fewer nodes and exponentially smaller depth than ReLU networks to approximate smooth functions to within a certain accuracy. This improved approximation power has practical consequences for large neural networks, given that a deep neural network is computationally expensive to train due to expensive gradient evaluations and slower convergence. The experiments conducted in Section 4 demonstrate the potential applications of these rational networks for solving PDEs and Generative Adversarial Networks (GANs).[1] The practical implementation of rational networks is straightforward in the TensorFlow framework and consists of replacing the activation functions by trainable rational functions. Finally, we highlight the main benefits of rational networks: the fast approximation of functions, the trainability of the activation parameters, and the smoothness of the activation function.

## 2  Rational neural networks

We consider neural networks whose activation functions consist of rational functions with trainable coefficients $a_i$ and $b_j$, i.e., functions of the form:

$$F(x) = \frac{P(x)}{Q(x)} = \frac{\sum_{i=0}^{r_P} a_i x^i}{\sum_{j=0}^{r_Q} b_j x^j}, \qquad a_P \neq 0, \quad b_Q \neq 0, \tag{1}$$

where $r_P$ and $r_Q$ are the polynomial degrees of the numerator and denominator, respectively. We say that $F(x)$ is of type $(r_P, r_Q)$ and degree $\max(r_P, r_Q)$.

The use of rational functions in deep learning is motivated by the theoretical work of Telgarsky, who proved error bounds on the approximation of ReLU neural networks by high-degree rational functions and vice versa [50]. On the practical side, neural networks based on rational activation functions are considered by Molina et al. [37], who defined a safe Padé Activation Unit (PAU) as

$$F(x) = \frac{\sum_{i=0}^{r_P} a_i x^i}{1 + |\sum_{j=1}^{r_Q} b_j x^j|}.$$

The denominator is selected so that $F(x)$ does not have poles located on the real axis. PAU networks can learn new activation functions and are competitive with state-of-the-art neural networks for image classification. However, this choice results in a non-smooth activation function and makes the gradient expensive to evaluate during training. In a closely related work, Chen et al. [8] propose high-degree rational activation functions in a neural network, which have benefits in terms of approximation power. However, this choice can significantly increase the number of parameters in the network, causing the training stage to be computationally expensive.

In this paper, we use low-degree rational functions as activation functions, which are then composed together by the neural network to build high-degree rational functions. In this way, we can leverage the approximation power of high-degree rational functions without making training expensive. We highlight the approximation power of rational networks and provide optimal error bounds to demonstrate that rational neural networks theoretically outperform ReLU networks. Motivated by our theoretical results, we consider rational activation functions of type $(3, 2)$, i.e., $r_P = 3$ and $r_Q = 2$. This type appears naturally in the theoretical analysis due to the composition property of Zolotarev sign functions (see Section 3.1): the degree of the overall rational function represented by the rational neural network is a whopping $3^{\#\text{layers}}$, while the number of trainable parameters only grows linearly with respect to the depth of the network. Moreover, a superdiagonal type $(3, 2)$ allows the rational activation function to behave like a nonconstant linear function at $\pm\infty$, unlike a diagonal type, e.g., $(2, 2)$, or the ReLU function. A low-degree activation function keeps the number of trainable parameters small, while the implicit composition in a neural network gives us the

approximation power of high-degree rationals. This choice is also motivated empirically, and we do not claim that the type $(3, 2)$ is the best choice for all situations as the configurations may depend on the application (see Figure 3 of the Supplementary Material). Our experiments on the approximation of smooth functions and GANs suggest that rational neural networks are an attractive alternative to ReLU networks (see Section 4). We observe that a good initialization, motivated by the theory of rational functions, prevents rational neural networks from having arbitrarily large values.

## 3 Theoretical results on rational neural networks

Here, we demonstrate the theoretical benefit of using neural networks based on rational activation functions due to their superiority over ReLU in approximating functions. We derive optimal bounds in terms of the total number of trainable parameters (also called size) needed by rational networks to approximate ReLU networks as well as functions in the Sobolev space $\mathcal{W}^{n,\infty}([0, 1]^d)$. Throughout this paper, we take $\epsilon$ to be a small parameter with $0 < \epsilon < 1$. We show that an $\epsilon$-approximation on the domain $[-1, 1]^d$ of a ReLU network by a rational neural network must have the following size (indicated in brackets):

$$\text{Rational } [\Omega(\log(\log(1/\epsilon)))] \leq \text{ReLU} \leq \text{Rational } [\mathcal{O}(\log(\log(1/\epsilon)))], \tag{2}$$

where the constants only depend on the size and depth of the ReLU network. Here, the upper bound means that all ReLU networks can be approximated to within $\epsilon$ by a rational network of size $\mathcal{O}(\log(\log(1/\epsilon)))$. The lower bound means that there is a ReLU network that cannot be $\epsilon$-approximated by a rational network of size less than $C \log(\log(1/\epsilon))$, for some constant $C > 0$. In comparison, the size needed by a ReLU network to approximate a rational neural network within the tolerance of $\epsilon$ is given by the following inequalities:

$$\text{ReLU } [\Omega(\log(1/\epsilon))] \leq \text{Rational} \leq \text{ReLU } [\mathcal{O}(\log(1/\epsilon))^3], \tag{3}$$

where the constants only depend on the size and depth of the rational neural network. This means that all rational networks can be approximated to within $\epsilon$ by a ReLU network of size $\mathcal{O}(\log(1/\epsilon))^3$, while there is a rational network that cannot be $\epsilon$-approximated by a ReLU network of size less than $\Omega(\log(1/\epsilon))$. A comparison between (2) and (3) suggests that rational networks could be more resourceful than ReLU. A key difference between rational networks and neural networks with polynomial activation functions is that polynomials perform poorly on non-smooth functions such as ReLU, with an algebraic convergence of $\mathcal{O}(1/\text{degree})$ [51] rather than the (root-)exponential convergence with rationals (see Figure 1 (left)).

### 3.1 Approximation of ReLU networks by rational neural networks

Telgarsky showed that neural networks and rational functions can approximate each other in the sense that there exists a rational function of degree[2] $\mathcal{O}(\text{polylog}(1/\epsilon))$ that is $\epsilon$-close to a ReLU network [50], where $\epsilon > 0$ is a small number. To prove this statement, Telgarsky used a rational function constructed with Newman polynomials [40] to obtain a rational approximation to the ReLU function that converges with square-root exponential accuracy. That is, Telgarsky needed a rational function of degree $\Omega(\log(1/\epsilon)^2)$ to achieve a tolerance of $\epsilon$. A degree $r$ rational function can be represented with $2(r + 1)$ coefficients, i.e., $a_0, \ldots, a_r$ and $b_0, \ldots, b_r$ in Equation (1). Therefore, the rational approximation to a ReLU network constructed by Telgarsky requires at least $\Omega(\text{polylog}(1/\epsilon))$ parameters. In contrast, for any rational function, Telgarsky showed that there exists a ReLU network of size $\mathcal{O}(\text{polylog}(1/\epsilon))$ that is an $\epsilon$-approximation on $[0, 1]^d$.

Our key observation is that by composing low-degree rational functions together, we can approximate a ReLU network much more efficiently in terms of the size (rather than the degree) of the rational network. Our theoretical work is based on a family of rationals called Zolotarev sign functions, which are the best rational approximation on $[-1, -\ell] \cup [\ell, 1]$, with $0 < \ell < 1$, to the sign function [3, 43], defined as

$$\text{sign}(x) = \begin{cases} -1, & x < 0, \\ 0, & x = 0, \\ 1, & x > 0. \end{cases}$$

A composition of $k \geq 1$ Zolotarev sign functions of type $(3, 2)$ has type $(3^k, 3^k - 1)$ but can be represented with $7k$ parameters instead of $2 \times 3^k + 1$. This property enables the construction of a rational approximation to ReLU using compositions of low-degree Zolotarev sign functions with $\mathcal{O}(\log(\log(1/\epsilon)))$ parameters in Lemma 1.

**Lemma 1** *Let $0 < \epsilon < 1$. There exists a rational network $R : [-1, 1] \to [-1, 1]$ of size $\mathcal{O}(\log(\log(1/\epsilon)))$ such that*

$$\|R - ReLU\|_\infty := \max_{x \in [-1,1]} |R(x) - ReLU(x)| \leq \epsilon.$$

*Moreover, no rational network of size smaller than $\Omega(\log(\log(1/\epsilon)))$ can achieve this.*

The proof of Lemma 1 (see Supplementary Material) shows that the given bound is optimal in the sense that a rational network requires at least $\Omega(\log(\log(1/\epsilon)))$ parameters to approximate the ReLU function on $[-1, 1]$ to within the tolerance $\epsilon > 0$. The convergence of the Zolotarev sign functions to the ReLU function is much faster, with respect to the number of parameters, than the rational constructed with Newman polynomials (see Figure 1 (left)).

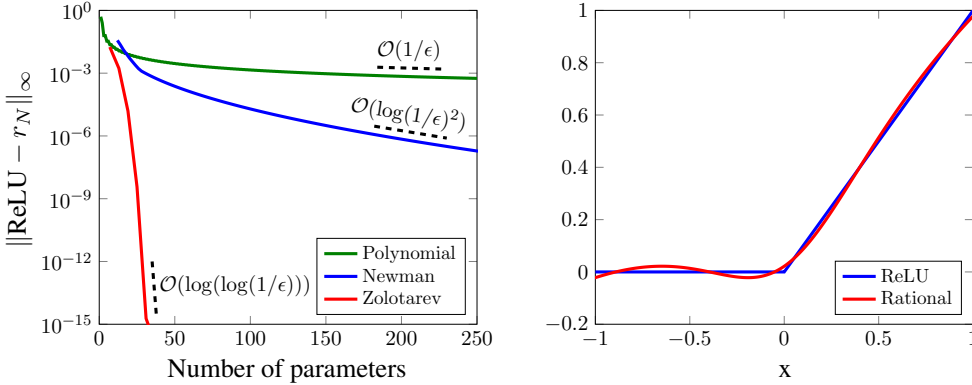

Figure 1: Left: Approximation error $\|\text{ReLU} - r_N\|_\infty$ of the Newman (blue), Zolotarev sign functions (red), and best polynomial approximation [42] of degree $N - 1$ (green) $r_N$ to ReLU with respect to the number of parameters required to represent $r_N$. Right: Best rational function of type $(3, 2)$ (red) that approximates the ReLU function (blue). We use this to initialize the rational activation functions when training a rational neural network.

The converse of Lemma 1, which is a consequence of a theorem proved by Telgarsky [50, Theorem 1.1], shows that any rational function can be approximated by a ReLU network of size at most $\mathcal{O}(\log(1/\epsilon)^3)$.

**Lemma 2** *Let $0 < \epsilon < 1$. If $R : [-1, 1] \to [-1, 1]$ is a rational function, then there exists a ReLU network $f : [-1, 1] \to [-1, 1]$ of size $\mathcal{O}(\log(1/\epsilon)^3)$ such that $\|R - f\|_\infty \leq \epsilon$.*

To demonstrate the improved approximation power of rational neural networks over ReLU networks ($\mathcal{O}(\log(\log(1/\epsilon)))$ versus $\mathcal{O}(\log(1/\epsilon)^3)$), it is known that a ReLU networks that approximates $x^2$, which is rational, to within $\epsilon$ on $[-1, 1]$ must be of size at least $\Omega(\log(1/\epsilon))$ [31, Theorem 11].

We can now state our main theorem based on Lemmas 1 and 2. Theorem 3 provides bounds on the approximation power of ReLU networks by rational neural networks and vice versa. We regard Theorem 3 as an analogue of [50, Theorem 1.1] for our Zolotarev sign functions, where we are counting the number of training parameters instead of the degree of the rational functions. In particular, our rational networks have high degrees but can be represented with few parameters due to compositions, making training more computationally efficient. While Telgarsky required a rational function with $\mathcal{O}(k^M \log(M/\epsilon)^M)$ parameters to approximate a ReLU network with fewer than $k$ nodes in each of $M$ layers to within a tolerance of $\epsilon$, we construct a rational network that only has size $\mathcal{O}(kM \log(\log(M/\epsilon)))$.

**Theorem 3** *Let $0 < \epsilon < 1$ and let $\| \cdot \|_1$ denote the vector 1-norm. The following two statements hold:*

*1. Let $R : [-1,1]^d \to [-1,1]$ be a rational network with $M$ layers and at most $k$ nodes per layer, where each node computes $x \mapsto r(a^\top x + b)$ and $r$ is a rational function with Lipschitz constant $L$ (a, b, and r are possibly distinct across nodes). Suppose further that $\|a\|_1 + |b| \leq 1$ and $r : [-1,1] \to [-1,1]$. Then, there exists a ReLU network $f : [-1,1]^d \to [-1,1]$ of size*

$$\mathcal{O}\left(kM \log(ML^M/\epsilon)^3\right)$$

*such that $\max_{x \in [-1,1]^d} |R(x) - f(x)| \leq \epsilon$.*

*2. Let $f : [-1,1]^d \to [-1,1]$ be a ReLU network with $M$ layers and at most $k$ nodes per layer, where each node computes $x \mapsto ReLU(a^\top x + b)$ and the pair $(a,b)$ (possibly distinct across nodes) satisfies $\|a\|_1 + |b| \leq 1$. Then, there exists a rational network $R : [-1,1]^d \to [-1,1]$ of size*

$$\mathcal{O}(kM \log(\log(M/\epsilon)))$$

*such that $\max_{x \in [-1,1]^d} |f(x) - R(x)| \leq \epsilon$.*

Theorem 3 highlights the improved approximation power of rational neural networks over ReLU networks. ReLU networks of size $\mathcal{O}(\mathrm{polylog}(1/\epsilon))$ are required to approximate rational networks while rational networks of size only $\mathcal{O}(\log(\log(1/\epsilon)))$ are sufficient to approximate ReLU networks.

## 3.2 Approximation of functions by rational networks

A popular question is the required size and depth of deep neural networks to approximate smooth functions [31, 38, 53]. In this section, we consider the approximation theory of rational networks. In particular, we consider the approximation of functions in the Sobolev space $\mathcal{W}^{n,\infty}([0,1]^d)$, where $n \geq 1$ is the regularity of the functions and $d \geq 1$. The norm of a function $f \in \mathcal{W}^{n,\infty}([0,1]^d)$ is defined as

$$\|f\|_{\mathcal{W}^{n,\infty}([0,1]^d)} = \max_{|\mathbf{n}| \leq n} \operatorname*{ess\,sup}_{\mathbf{x} \in [0,1]^d} |D^{\mathbf{n}} f(\mathbf{x})|,$$

where $\mathbf{n}$ is the multi-index $\mathbf{n} = (n_1, \ldots, n_d) \in \{0, \ldots, n\}^d$, and $D^{\mathbf{n}} f$ is the corresponding weak derivative of $f$. In this section, we consider the approximation of functions from

$$F_{d,n} := \{f \in \mathcal{W}^{n,\infty}([0,1]^d), \quad \|f\|_{\mathcal{W}^{n,\infty}([0,1]^d)} \leq 1\}.$$

By the Sobolev embedding theorem [7], this space contains the functions in $\mathcal{C}^{n-1}([0,1]^d)$, which is the class of functions whose first $n-1$ derivatives are Lipschitz continuous. Yarotsky derived upper bounds on the size of neural networks with piecewise linear activation functions needed to approximate functions in $F_{d,n}$ [53, Theorem 1]. In particular, he constructed an $\epsilon$-approximation to functions in $F_{d,n}$ with a ReLU network of size at most $\mathcal{O}(\epsilon^{-d/n} \log(1/\epsilon))$ and depth smaller than $\mathcal{O}(\log(1/\epsilon))$. The term $\epsilon^{-d/n}$ is introduced by a local Taylor approximation, while the $\log(1/\epsilon)$ term is the size of the ReLU network needed to approximate monomials, i.e., $x^j$ for $j \geq 0$, in the Taylor series expansion.

We now present an analogue of Yarotsky's theorem for a rational neural network.

**Theorem 4** *Let $d \geq 1$, $n \geq 1$, $0 < \epsilon < 1$, and $f \in F_{d,n}$. There exists a rational neural network $R$ of size*

$$\mathcal{O}(\epsilon^{-d/n} \log(\log(1/\epsilon)))$$

*and maximum depth $\mathcal{O}(\log(\log(1/\epsilon)))$ such that $\|f - R\|_\infty \leq \epsilon$.*

The proof of Theorem 4 consists of approximating $f$ by a local Taylor expansion. One needs to approximate the piecewise linear functions and monomials arising in the Taylor expansion by rational networks using Lemma 1 and Proposition 6 (see Supplementary Material). The main distinction between Yarotsky's argument and the proof of Theorem 4 is that monomials can be represented by rational neural networks with a size that does not depend on the accuracy of $\epsilon$. In contrast, ReLU networks require $\mathcal{O}(\log(1/\epsilon))$ parameters. Meanwhile, while ReLU neural networks can exactly approximate piecewise linear functions with a constant number of parameters, rational networks can approximate them with a size of a most $\mathcal{O}(\log(\log(1/\epsilon)))$ (see Lemma 1). That is, rational neural networks approximate piecewise linear functions much faster than ReLU networks approximate polynomials. This allows the existence of a rational network approximation to $f$ with exponentially smaller depth ($\mathcal{O}(\log(\log(1/\epsilon)))$) than the ReLU networks constructed by Yarotsky.

A theorem proved by DeVore et al. [13] gives a lower bound of $\Omega(\epsilon^{-d/n})$ on the number of parameters needed by a neural network to express any function in $F_{d,n}$ with an error $\epsilon$, under the assumption that the weights are chosen continuously. Comparing $\mathcal{O}(\epsilon^{-d/n} \log(\log(1/\epsilon)))$ and $\mathcal{O}(\epsilon^{-d/n} \log(1/\epsilon))$, we find that rational neural networks require exponentially fewer nodes than ReLU networks with respect to the optimal bound of $\Omega(\epsilon^{-d/n})$ to approximate functions in $F_{d,n}$.

## 4    Experiments using rational neural networks

In this section, we consider neural networks with trainable rational activation functions of type $(3, 2)$. We select the type $(3, 2)$ based on empirical performance; roughly, a low-degree (but higher than 1) rational function is ideal for generating high-degree rational functions by composition, with a small number of parameters. The rational activation units can be easily implemented in the open-source TensorFlow library [2] by using the `polyval` and `divide` commands for function evaluations. The coefficients of the numerators and denominators of the rational activation functions are trainable parameters, determined at the same time as the weights and biases of the neural network by backpropagation and a gradient descent optimization algorithm.

One crucial question is the initialization of the coefficients of the rational activation functions [8, 37]. A badly initialized rational function might contain poles on the real axis, leading to exploding values, or converge to a local minimum in the optimization process. Our experiments, supported by the empirical results of Molina et al. [37], show that initializing each rational function with the best rational approximation to the ReLU function (as described in Lemma 1) produces good performance. The underlying idea is to initialize rational networks near a network with ReLU activation functions, widely used for deep learning. Then, the adaptivity of the rational functions allows for further improvements during the training phase. We represent the initial rational function used in our experiments in Figure 1 (right). The coefficients of this function are obtained by using the `minimax` command, available in the Chebfun software [14, 15] for numerically computing rational approximations (see Table 1 in the Supplementary Material).

In the following experiments, we use a single rational activation function of type $(3, 2)$ at each layer, instead of different functions at each node to reduce the number of trainable parameters and the computational training expense.

### 4.1    Approximation of functions

Raissi, Perdikaris, and Karniadakis [45, 46] introduce a framework called *deep hidden physics models* for discovering nonlinear partial differential equations (PDEs) from observations. This technique requires to solving the following interpolation problem: given the observation data $(u_i)_{1 \leq i \leq N}$ at the spatio-temporal points $(x_i, t_i)_{1 \leq i \leq N}$, find a neural network $\mathcal{N}$ (called the identification network), that minimizes the loss function

$$\mathcal{L} = \frac{1}{N} \sum_{i=1}^{N} |\mathcal{N}(x_i, t_i) - u_i|^2. \tag{4}$$

This technique has successfully discovered hidden models in fluid mechanics [47], solid mechanics [21], and nonlinear partial differential equations such as the Korteweg–de Vries (KdV) equation [46]. Raissi et al. use an identification network, consisting of 4 layers and 50 nodes per layer, to interpolate samples from a solution to the KdV equation. Moreover, they observe that networks based on smooth activation functions, such as the hyperbolic tangent $(\tanh(x))$ or the sinusoid $(\sin(x))$, outperform ReLU neural networks [45, 46]. However, the performance of these smooth activation functions highly depends on the application.

Moreover, these functions might not be adapted to approximate non-smooth or highly oscillatory solutions. Recently, Jagtap, Kawaguchi, and Karnidakis [25] proposed and analyzed different adaptive activation functions to approximate smooth and discontinuous functions with physics-informed neural networks. More specifically, they use an adaptive version of classical activation functions such as sigmoid, hyperbolic tangent, ReLU, and Leaky ReLU. The choice of these trainable activation functions introduces another parameter in the design of the neural network architecture, which may not be ideal for use for a black-box data-driven PDE solver.

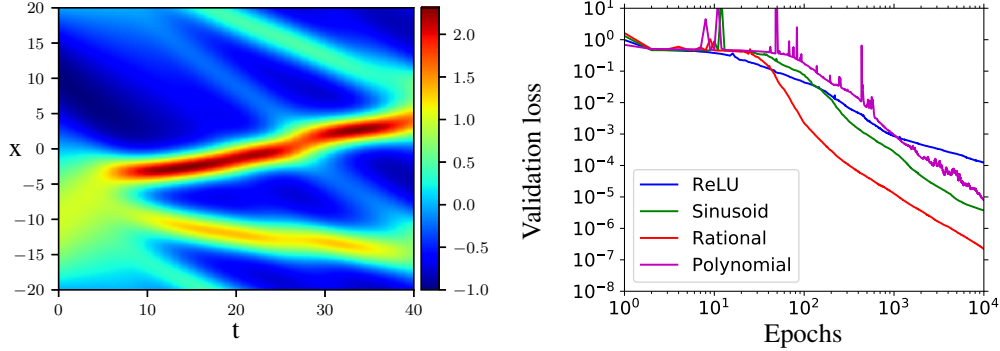

Figure 2: Solution to the KdV equation used as training data (left) and validation loss of a ReLU (blue), sinusoid (green), rational (red), and polynomial (purple) neural networks with respect to the number of optimization steps (right).

We illustrate that rational neural networks can address the issues mentioned above due to their adaptivity and approximation power (see Section 3). Similarly to Raissi [45], we use a solution $u$ to the KdV equation:

$$u_t = -uu_x - u_{xxx}, \quad u(x,0) = -\sin(\pi x/20),$$

as training data for the identification network (see the left panel of Figure 2). We train and compare four neural networks, which contain ReLU, sinusoid, rational, and polynomial activation functions, respectively.[3] The mean squared error (MSE) of the neural networks on the validation set throughout the training phase is reported in the right panel of Figure 2. We observe that the rational neural network outperforms the sinusoid network, despite having the same asymptotic convergence rate. The network with polynomial activation functions (chosen to be of degree 3 in this example) is harder to train than the rational network, as shown by the non-smooth validation loss (see the right panel of Figure 2). We highlight that rational neural networks are never much bigger in terms of trainable parameters than ReLU networks since the increase is only linear with respect to the number of layers. Here, the ReLU network has 8000 parameters (consisting of weights and biases), while the rational network has $8000 + 7 \times \#\text{layers} = 8035$. The ReLU, sinusoid, rational, and polynomial networks achieve the following mean square errors after $10^4$ epochs:

$$\text{MSE}(u_{\text{ReLU}}) = 1.9 \times 10^{-4}, \qquad \text{MSE}(u_{\text{sin}}) = 3.3 \times 10^{-6},$$
$$\text{MSE}(u_{\text{rat}}) = 1.2 \times 10^{-7}, \qquad \text{MSE}(u_{\text{poly}}) = 3.6 \times 10^{-5}.$$

The absolute approximation errors between the different neural networks and the exact solution to the KdV equation is illustrated in Figure 2 of the Supplementary Material. The rational neural network is approximatively five times more accurate than the sinusoid network used by Raissi and twenty times more accurate than the ReLU network. Moreover, the approximation errors made by the ReLU network are not uniformly distributed in space and time and located in specific regions, indicating that a network with non-smooth activation functions is not appropriate to resolve smooth solutions to PDEs.

## 4.2 Generative adversarial networks

Generative adversarial networks (GANs) are used to generate fake examples from an existing dataset [18]. They usually consist of two networks: a generator to produce fake samples and a discriminator to evaluate the samples of the generator with the training dataset. Radford et al. [44] describe deep convolutional generative adversarial networks (DCGANs) to build good image representations using convolutional architectures. They evaluate their model on the MNIST and Imagenet image datasets [12, 30]. This section highlights the simplicity of using rational activation functions in existing neural network architectures by training an Auxiliary Classifier GAN (ACGAN) [41] on the MNIST dataset. In particular, the neural network[4], denoted by ReLU network

in this section, consists of convolutional generator and discriminator networks with ReLU and Leaky ReLU [34] activation units (respectively) and is used as a reference GAN. As in the experiment described in Section 4.1, we replace the activation units of the generative and discriminator networks by a rational function with trainable coefficients (see Figure 1). We initialize the activation functions in the training phase with the best rational function that approximates the ReLU function on $[-1, 1]$.

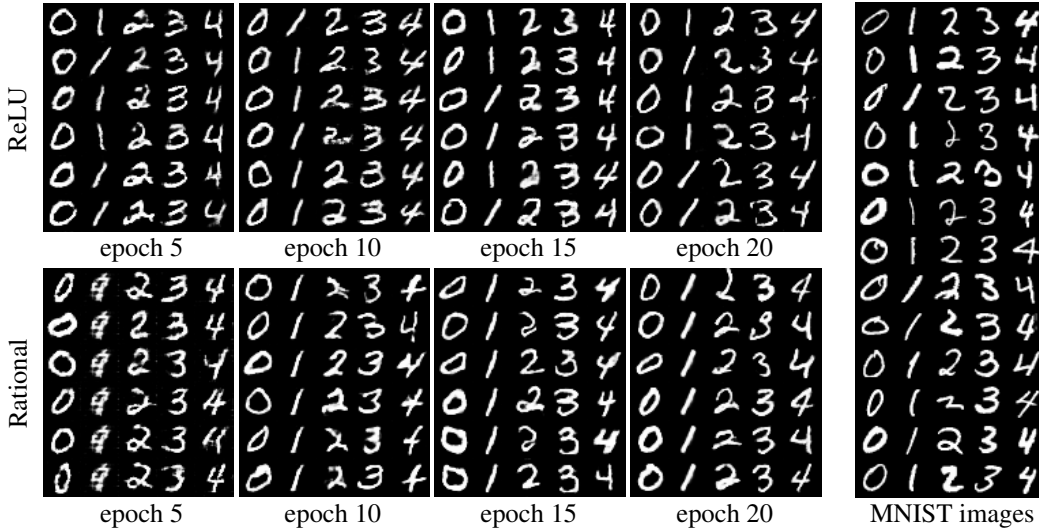

Figure 3: Digits generated by a ReLU (top) and rational (bottom) auxiliary classifier generative adversarial network. The right panel contains samples from the first five classes of the MNIST dataset for comparison.

We show images of digits from the first five classes generated by a ReLU and rational GANs at different epochs of the training in Figure 3 (the samples are generated randomly and are not manually selected). We observe that a rational network can generate realistic images with a broader range of features than the ReLU network, as illustrated by the presence of bold numbers at the epoch 20 in the bottom panel of Figure 3. However, the digits one generated by the rational network are identical, suggesting that the rational GAN suffers from mode collapse. It should be noted that generative adversarial networks are notoriously tricky to train [17]. The hyper-parameters of the reference model are intensively tuned for a piecewise linear activation function (as shown by the use of Leaky ReLU in the discriminator network). Moreover, many stabilization methods have been proposed to resolve the mode collapse and non-convergence issues in training, such as Wasserstein GAN [4], Unrolled Generative Adversarial Networks [35], and batch normalization [24]. These techniques could be explored and combined with rational networks to address the mode collapse issue observed in this experiment.

## 5    Conclusions

We have investigated rational neural networks, which are neural networks with smooth trainable activation functions based on rational functions. Theoretical statements demonstrate the improved approximation power of rational networks in comparison with ReLU networks. In practice, it seems beneficial to select the activation function as very low-degree rationals, making training more computationally efficient. We emphasize that it is simple to implement rational networks in existing deep learning architectures, such as TensorFlow, together with the ability to have trainable activation functions.

There are many future research directions exploring the potential applications of rational networks in fields such as image classification, time series forecasting, and generative adversarial networks. These applications already employ nonstandard activation functions to overcome various drawbacks of ReLU. Another exciting and promising field is the numerical solution and data-driven discovery of partial differential equations with deep learning. We believe that popular techniques such as

physics-informed neural networks [46] could benefit from rational neural networks to improve the robustness and performances of PDE solvers, both from a theoretical and practical viewpoint.

## Broader Impact

Neural networks have applications in diverse fields such as facial recognition, credit-card fraud, speech recognition, and medical diagnosis. There is a growing understanding of the approximation power of neural networks, which is adding theoretical justification to their use in societal applications. We are particularly interested in the future applicability of rational neural networks in discovering and solving of partial differential equations (PDEs). Neural networks, in particular rational neural networks, have the potential to revolutionize fields where PDE models derived by mechanistic principles are lacking.

## Acknowledgments and Disclosure of Funding

The authors thank the National Institute of Informatics (Japan) for funding a research visit, during which this project was initiated. We thank Gilbert Strang for making us aware of Telgarsky's paper [50]. We also thank Matthew Colbrook and Nick Trefethen for their suggestions on the paper. This work is supported by the EPSRC Centre For Doctoral Training in Industrially Focused Mathematical Modelling (EP/L015803/1) in collaboration with Simula Research Laboratory. The work of the third author is supported by the National Science Foundation grant no. 1818757.

## Footnotes

[1]All code and hyper-parameters are publicly available at [6].

[2]A polylogarithmic function in $x$ is any polynomial in $\log(x)$ and is denoted by $\text{polylog}(x)$.

[3]Details of the parameters used for this experiment are available in the Supplementary Material.

[4]We use the TensorFlow implementation available at [1] and provide extended details and results of the experiment in the Supplementary Material.

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
