[Supplementary Material]

# Supplementary Material of Rational neural networks

**Nicolas Boullé**
Mathematical Institute
University of Oxford
Oxford, OX2 6GG, UK
boulle@maths.ox.ac.uk

**Yuji Nakatsukasa**
Mathematical Institute
University of Oxford
Oxford, OX2 6GG, UK
nakatsukasa@maths.ox.ac.uk

**Alex Townsend**
Department of Mathematics
Cornell University
Ithaca, NY 14853, USA
townsend@cornell.edu

## A    Deferred proofs of Section 3.1

We first show that a rational function can approximate the absolute value function $|x|$ on $[-1, 1]$ with square-root exponential convergence.

**Lemma 1** *For any integer $k \geq 0$, we have*

$$\min_{r \in \mathcal{R}_{k,k}} \max_{x \in [-1,1]} ||x| - xr(x)| \leq 4e^{-\pi\sqrt{k/2}},$$

*where $\mathcal{R}_{k,k}$ is the space of rational functions of type at most $(k, k)$. Thus, $xr(x)$ is a rational approximant to $|x|$ of type at most $(k + 1, k)$.*

*Moreover, if $k = \prod_{i=1}^{p} k_i$ for some $p \geq 1$ and integers $k_1, \dots, k_p \geq 2$, then $r$ can be written as $r = R_p \circ \cdots \circ R_1$, where $R_i \in \mathcal{R}_{k_i,k_i}$.*

**Proof.** Let $0 < \ell < 1$ be a real number and consider the sign function on the domain $[-1, -\ell] \cup [\ell, 1]$, i.e.,

$$\text{sign}(x) = \begin{cases} -1, & x \in [-1, -\ell], \\ +1, & x \in [\ell, 1]. \end{cases}$$

By [2, Equation (33)], we find that for any $k \geq 0$,

$$\min_{r \in \mathcal{R}_{k,k}} \max_{x \in [-1,-\ell] \cup [\ell,1]} |\text{sign}(x) - r(x)| \leq 4 \left[ \exp\left( \frac{\pi^2}{2\log(4/\ell)} \right) \right]^{-k}.$$

Let $r(x)$ be the rational function of type $(k, k)$ that attains the minimum [2, Equation (12)]. We refer to such $r(x)$ as the Zolotarev sign function. It is given by

$$r(x) = Mx \frac{\prod_{j=1}^{\lfloor (k-1)/2 \rfloor} x^2 + c_{2j}}{\prod_{j=1}^{\lfloor k/2 \rfloor} x^2 + c_{2j-1}}, \quad c_j = \ell^2 \frac{\text{sn}^2(jK(\kappa)/k; \kappa)}{1 - \text{sn}^2(jK(\kappa)/k; \kappa)}.$$

Here, $M$ is a real constant selected so that $\text{sign}(x) - r(x)$ equioscillates on $[-1, -\ell] \cup [\ell, 1]$, $\kappa = \sqrt{1 - \ell^2}$, $\text{sn}(\cdot)$ is the first Jacobian elliptic function, and $K$ is the complete elliptic integral of the first kind. Since $|x| = x \cdot \text{sign}(x)$ we have the following inequality,

$$\max_{x \in [-1,-\ell] \cup [\ell,1]} ||x| - xr(x)| = \max_{x \in [-1,-\ell] \cup [\ell,1]} |x \cdot \text{sign}(x) - xr(x)|$$
$$\leq \max_{x \in [-1,-\ell] \cup [\ell,1]} |\text{sign}(x) - r(x)|.$$

The last inequality follows because $|x| \leq 1$ on $[-1, -\ell] \cup [\ell, 1]$. Moreover, since $xr(x) \geq 0$ for $x \in [-1, 1]$ (see [2, Equation (12)]) we have

$$\max_{x \in [-\ell,\ell]} ||x| - xr(x)| \leq \max_{x \in [-\ell,\ell]} |x| \leq \ell.$$

Therefore,

$$\max_{x \in [-1,1]} ||x| - xr(x)| \le \max \left\{ \ell, 4 \left[ \exp \left( \frac{\pi^2}{2 \log(4/\ell)} \right) \right]^{-k} \right\}.$$

Now, select $0 < \ell < 1$ to minimize this upper bound. One finds that $\ell = 4 \exp(-\pi \sqrt{k/2})$ and the result follows immediately.

For the final claim, let $r$ be the Zolotarev sign function $Z_k(\cdot\,;\ell)$ of type $(k, k)$ on $[-1, -\ell] \cup [\ell, 1]$, with $k = \prod_{i=1}^p k_i$. By definition, $Z_k(\cdot\,;\ell)$ is the best rational approximation of degree $k$ to the sign function on $[-1, -\ell] \cup [\ell, 1]$. We know from [7, 8] that there exist $p$ Zolotarev sign functions $R_1, \ldots, R_p$, where each $R_i$ is of type $(k_i, k_i)$, such that

$$r(x) := Z_k(x; \ell) = R_p(\cdots (R_2(R_1(x))) \cdots). \tag{1}$$

∎

The proof of Lemma 1 is a direct consequence of the previous lemma and the properties of Zolotarev sign functions.

**Proof of Lemma 1.** Let $0 < \epsilon < 1$, $0 < \ell < 1$, $k \ge 1$, and $r$ be the Zolotarev sign function $Z_{3^k}(\cdot\,;\ell)$ of type $(3^k, 3^k - 1)$. Again from [7, 8], we see that there exist $k$ Zolotarev sign functions $R_1, \ldots, R_k$ of type $(3, 2)$ such that their composition equals $Z_{3^k}(x; \ell)$, i.e.,

$$r(x) := Z_{3^k}(x; \ell) = R_k(\cdots (R_2(R_1(x))) \cdots). \tag{2}$$

Following the proof of Lemma 1, we have the inequality

$$\max_{x \in [-1,1]} ||x| - xr(x)| \le 4e^{-\pi \sqrt{3^k/2}}, \tag{3}$$

where we chose $\ell = 4 \exp(-\pi \sqrt{3^k/2})$. Now, we take

$$k = \left\lceil \frac{\ln(2/\pi^2) + 2 \ln(\ln(4/\epsilon))}{\ln(3)} \right\rceil, \tag{4}$$

so that the right-hand side of Equation (3) is bounded by $\epsilon$. Finally, we use the identity

$$\text{ReLU}(x) = \frac{|x| + x}{2}, \quad x \in \mathbb{R},$$

to define a rational approximation to the ReLU function on the interval $[-1, 1]$ as

$$\tilde{r}(x) = \frac{1}{2} \left( \frac{xr(x)}{1 + \epsilon} + x \right).$$

Therefore, we have the following inequalities for $x \in [-1, 1]$,

$$|\text{ReLU}(x) - \tilde{r}(x)| = \frac{1}{2} \left| |x| - \frac{xr(x)}{1 + \epsilon} \right| \le \frac{1}{2(1 + \epsilon)} (||x| - xr(x)| + \epsilon |x|)$$

$$\le \frac{\epsilon}{1 + \epsilon} \le \epsilon.$$

Then, $r$ is a composition of $k$ rational functions of type $(3, 2)$ and can be represented using at most $7k$ coefficients (see Equation (1)). Moreover, using Equation (4), we see that $k = \mathcal{O}(\log(\log(1/\epsilon)))$, which means that $\tilde{r}$ is representable by a rational network of size $\mathcal{O}(\log(\log(1/\epsilon)))$. Finally, $|\tilde{r}(x)| \le 1$ for $x \in [-1, 1]$. ∎

The upper bound on the complexity of the neural network obtained in Lemma 1 is optimal, as proved by Vyacheslavov [13].

**Theorem 2 (Vyacheslavov)** *The following inequalities hold:*

$$C_1 e^{-\pi \sqrt{k}} \le \max_{x \in [-1,1]} ||x| - r_k(x)| \le C_2 e^{-\pi \sqrt{k}}, \qquad k \ge 0, \tag{5}$$

*where $r_k$ is the best rational approximation to $|x|$ in $[-1, 1]$ from $\mathcal{R}_{k,k}$. Here, $C_1, C_2 > 0$ are constants that are independent of $k$.*

We first deduce the following corollary, giving lower and upper bounds on the optimal rational approximation to the ReLU function.

**Corollary 3** *The following inequalities hold:*

$$\frac{C_1}{2}e^{-\pi\sqrt{k}} \leq \|ReLU - r_k\|_\infty \leq \frac{C_2}{2}e^{-\pi\sqrt{k}}, \qquad k \geq 0, \tag{6}$$

*where $r_k$ is the best rational approximation to ReLU on $[-1, 1]$ in $\mathcal{R}_{k,k}$ and $C_1, C_2 > 0$ are constants given by Theorem 2.*

**Proof.** Let $k$ be an integer and let $r_k \in \mathcal{R}_{k,k}$ be any rational function of degree $\leq k$. Now, define $r_{\text{abs}}(x) = 2r_k(x) - x$. Since $\text{ReLU}(x) = (|x| + x)/2$, we have

$$\|\text{ReLU}-r_k\|_\infty = \max_{x\in[-1,1]}\left|\frac{1}{2}(r_{\text{abs}}(x) + x) - \frac{1}{2}(|x| + x)\right| = \max_{x\in[-1,1]}\frac{1}{2}|r_{\text{abs}}(x) - |x|| \geq \frac{1}{2}C_1 e^{-\pi\sqrt{k}},$$

where the inequality is from Theorem 2. Now, let $r_k \in \mathcal{R}_{k,k}$ be the best rational approximation to $|x|$ on $[-1, 1]$. Now, define $r_{\text{ReLU}}(x) = (r_k(x) + x)/2$. We find that

$$\|\text{ReLU}-r_{\text{ReLU}}\|_\infty = \max_{x\in[-1,1]}\left|\frac{1}{2}(|x| + x) - \frac{1}{2}(r_k(x) + x)\right| = \max_{x\in[-1,1]}\frac{1}{2}||x| - r_k(x)| \leq \frac{1}{2}C_2 e^{-\pi\sqrt{k}},$$

which proves that the best approximation to ReLU satisfies the upper bound. ∎

We now show that a rational neural network must be at least $\Omega(\log(\log(1/\epsilon)))$ in size (total number of nodes) to approximate the ReLU function to within $\epsilon$.

**Proposition 4** *Let $0 < \epsilon < 1$. A rational neural network that approximates the ReLU function on $[-1, 1]$ to within $\epsilon$ has size of at least $\Omega(\log(\log(1/\epsilon)))$.*

**Proof.** Let $R : [-1, 1] \to \mathbb{R}$ be a rational neural network with $k_1, \ldots, k_M \geq 1$ nodes at each of its $M$ layers, and assume that its activation functions are rational functions of type at most $(r_P, r_Q)$. Let $d_r = \max(r_P, r_Q)$ be the maximum of the degrees of the activation functions of $R$. Such a network has size $\sum_{i=1}^M k_i$. Note that $R$ itself is a rational function of degree $d$, where from additions and compositions of rational functions we have $d \leq d_r^M \prod_{i=1}^M k_i$. If $R$ is an $\epsilon$-approximation to the ReLU function on $[-1, 1]$, we know by Corollary 3 that

$$\frac{C_1}{2}e^{-\pi\sqrt{d}} \geq \epsilon, \qquad d \geq \left(\frac{1}{\pi}\ln\left(\frac{C_1}{2\epsilon}\right)\right)^2. \tag{7}$$

The statement follows by minimizing the size of $R$, i.e., $\sum_{i=1}^M k_i$ subject to

$$d_r^M \prod_{i=1}^M k_i \geq \left(\frac{1}{\pi}\ln\left(\frac{C_1}{2\epsilon}\right)\right)^2.$$

That is,

$$\sum_{i=1}^M \ln(k_i) + M\ln(d_r) \geq 2\ln\left(\ln\left(\frac{C_1}{2\epsilon}\right)\right) - 2\ln(\pi). \tag{8}$$

We introduce a Lagrange multiplier $\lambda \in \mathbb{R}$ and define the Lagrangian of this optimization problem as

$$\mathcal{L}(k_1, \ldots, k_M, \lambda) = \sum_{i=1}^M k_i + \lambda\left[2\ln\left(\ln\left(\frac{C_1}{2\epsilon}\right)\right) - 2\ln(\pi) - \sum_{i=1}^M \ln(k_i) - M\ln(d_r)\right].$$

One finds using the Karush–Kuhn–Tucker conditions [6] that $k_1 = \cdots = k_M = \lambda$. Then, using Equation (8), we find that $\lambda$ satisfies

$$\ln(\lambda) \geq \frac{2}{M}\left[\ln\left(\ln\left(\frac{C_1}{2\epsilon}\right)\right) - \ln(\pi)\right] - \ln(d_r) =: \ln(\lambda^*). \tag{9}$$

Therefore, the rational network $R$ with $M$ layers that approximates the ReLU function to within $\epsilon$ on $[-1, 1]$ has a size of at least $s(M) := M\lambda^*$, where $\lambda^*$ is given by Equation (9) and depends on $M$.

We now minimize $s(M)$ with respect to the number of layers $M \geq 1$. We remark that minimizing $s$ is equivalent of minimizing $\ln(s)$, where

$$\ln(s(M)) = \ln(M) + \ln(\lambda^*) = \ln(M) + \frac{2}{M}\left[\ln\left(\ln\left(\frac{C_1}{2\epsilon}\right)\right) - \ln(\pi)\right] - \ln(d_r).$$

One finds that one should take $k_1 = \cdots = k_M = \lambda^* = \mathcal{O}(1)$ and $M = \Omega(\log(\log(1/\epsilon)))$. The result follows. $\blacksquare$

We now show that ReLU neural networks can approximate rational functions.

**Proof of Lemma 2.** Let $0 < \epsilon < 1$ and $R : [-1, 1] \to [-1, 1]$ be a rational function. Take $\tilde{R}(x) = R(2x - 1)$, which is still a rational function. Without loss of generality, we can assume that $\tilde{R}$ is an irreducible rational function (otherwise cancel factors till it is irreducible). Since $\tilde{R}$ is a rational, it can be written as $\tilde{R} = p/q$ with $\max_{x \in [0,1]} |q(x)| = 1$. Moreover, we know that $\tilde{R}(x) \in [-1, 1]$ for $x \in [0, 1]$ so we can assume that $q(x) \geq 0$ for $x \in [0, 1]$ (it is either positive or negative by continuity). Since $R$ is continuous on $[-1, 1]$, there is an integer $n \geq 1$ such that $q(x) \in [2^{-n}, 1]$ for $x \in [0, 1]$. Furthermore, we find that $|p(x)| \leq 1$ for $x \in [0, 1]$ because $|R(x)| \leq 1$ and $|q(x)| \leq 1$ for $x \in [0, 1]$. By [12, Theorem 1.1], there exists a ReLU network $f : [0, 1] \to \mathbb{R}$ of size $\mathcal{O}(n^7 \log(1/\epsilon)^3)$ such that

$$\max_{x \in [0,1]} \left| f(x) - \frac{p(x)}{q(x)} \right| \leq \frac{\epsilon}{2}.$$

We now define a scaled ReLU network $\tilde{f}(x) = f(x)/(1 + \epsilon/2)$ such that $|\tilde{f}(x)| \leq 1$ for $x \in [0, 1]$. Therefore, for all $x \in [0, 1]$,

$$\left| \tilde{f}(x) - \tilde{R}(x) \right| = \left| \frac{f(x)}{1 + \epsilon/2} - \frac{p(x)}{q(x)} \right| \leq \frac{1}{1 + \epsilon/2} \left( \left| f(x) - \frac{p(x)}{q(x)} \right| + \frac{\epsilon}{2} \left| \frac{p(x)}{q(x)} \right| \right) \leq \epsilon.$$

Therefore, $x \mapsto \tilde{f}((x + 1)/2)$ is a ReLU neural network of size $\mathcal{O}(\log(1/\epsilon)^3)$ that is an $\epsilon$-approximation to $R$ on $[-1, 1]$. $\blacksquare$

We can now prove Theorem 3 that shows how rational neural networks can approximate ReLU networks and vice versa. The structure of the proof closely follows [12, Lemma 1.3].

**Proof of Theorem 3.** The statement of Theorem 3 comes in two parts, and we prove them separately.
1. Consider the subnetwork $H$ of the rational network $R$, consisting of the layers of $R$ up to the $J$th layer for some $1 \leq J \leq M - 1$. Let $H_{\text{ReLU}}$ denote the ReLU network obtained by replacing each rational function $r_{ij}$ in $H$ by a ReLU network approximation $f_{r_{ij}}$ at a given tolerance $\epsilon_j > 0$ for $1 \leq j \leq J$ and $1 \leq i \leq k_j$, such that $|H_{\text{ReLU}}(x)| \leq 1$ for $x \in [-1, 1]$ (see Lemma 2). Let $x \mapsto r_{i,J+1}(a_{i,J+1}^\top H(x) + b_{i,J+1})$ be the output of the rational network $R$ at layer $J + 1$ and node $i$ for $1 \leq i \leq k_J$. Now, approximate node $i$ in the $(J + 1)$st layer by a ReLU network $f_{r_i,J+1}$ with tolerance $\epsilon_{J+1} > 0$ (see Lemma 2). The approximation error $E_{i,J+1}$ between the rational and the approximating ReLU network at layer $J + 1$ and node $i$ satisfies

$$E_{i,J+1} = |f_{r_{i,J+1}}(a_{i,J+1}^\top H_{\text{ReLU}}(x) + b_{i,J+1}) - r_{i,J+1}(a_{i,J+1}^\top H(x) + b_{i,J+1})|$$
$$\leq \underbrace{|f_{r_{i,J+1}}(a_{i,J+1}^\top H_{\text{ReLU}}(x) + b_{i,J+1}) - r_{i,J+1}(a_{i,J+1}^\top H_{\text{ReLU}}(x) + b_{i,J+1})|}_{(1)}$$
$$+ \underbrace{|r_{i,J+1}(a_{i,J+1}^\top H_{\text{ReLU}}(x) + b_{i,J+1}) - r_{i,J+1}(a_{i,J+1}^\top H(x) + b_{i,J+1})|}_{(2)}.$$

The first term is bounded by

$$(1) \leq \max_{x \in [-1,1]} |r_{i,J+1}(x) - f_{r_{i,J+1}}| \leq \epsilon_{J+1},$$

since $|a_{i,J+1}^\top H_{\text{ReLU}}(x) + b_{i,J+1}| \leq \|a_{i,J+1}\|_1 + |b_{i,J+1}| \leq 1$ by assumption. The second term is bounded as the Lipschitz constant of $r_{i,J+1}$ is at most $L$. That is,

$$(2) \leq L\|a_{i,J+1}\|_1 \max_{x \in [-1,1]^d} \|H_{\text{ReLU}}(x) - H(x)\|_\infty \leq L \max_{x \in [-1,1]^d} \|H_{\text{ReLU}}(x) - H(x)\|_\infty,$$

where we used the fact that $\|a_{i,J+1}\|_1 \leq 1$ and $\|H_{\text{ReLU}}(x)\|_\infty \leq 1$ for $x \in [-1,1]^d$. We find that we have the following set of inequalities:

$$\max_{1 \leq i \leq k_{j+1}} E_{i,j+1} \leq L \max_{1 \leq i \leq k_j} E_{i,j} + \epsilon_{j+1}, \qquad 1 \leq i \leq k_j, \quad 1 \leq j \leq J+1,$$

with $E_{i,0} = 0$. If we select $\epsilon_j = \epsilon L^{j-J-1}/(J+1)$, then we find that $\max_{1 \leq i \leq k_{J+1}} E_{i,J+1} \leq \epsilon$. When $J = M - 1$, the ReLU network approximates the original rational network, $R$, and the ReLU network has size

$$\mathcal{O}\left( k \sum_{j=1}^{M} \log\left( \frac{M}{L^{j-M}\epsilon} \right)^3 \right).$$

where we used the fact that $k_j \leq k$ for $1 \leq j \leq M$. This can be simplified a little since

$$\sum_{j=1}^{M} \log\left( \frac{M}{L^{j-M}\epsilon} \right)^3 = \sum_{j=1}^{M} \left( \log(ML^M/\epsilon) + j\log(1/L) \right)^3 = \mathcal{O}\left( M \log(ML^M/\epsilon)^3 \right).$$

2. Telgarsky proved in [12, Lemma 1.3] that if $H_R$ is a neural network obtained by replacing all the ReLU activation functions in $f$ by rational functions $R$ for $1 \leq j \leq M$, which satisfies $R(x) \in [-1,1]$ and $|R(x) - \text{ReLU}(x)| \leq \epsilon/M$ for $x \in [-1,1]$, then

$$\max_{x \in [-1,1]^d} |f(x) - H_R(x)| \leq \epsilon.$$

Let $R$ be a rational neural network approximating ReLU with a tolerance of $\epsilon/M$, constructed by Lemma 1. Then, $R$ is rational network of size $\mathcal{O}(\log(\log(M/\epsilon)))$ and thus, $H_R$ is a rational neural network of size $\mathcal{O}(Mk\log(\log(M/\epsilon)))$. ∎

## B  Deferred proofs of Section 3.2

Here, we show that the construction in Lemma 1 can approximate any piecewise linear function on $[-1,1]$.

**Proposition 5** *Let $0 < \epsilon < 1$ and let $g : [0,1] \to \mathbb{R}$ be any continuous piecewise linear function with $m \geq 1$ breakpoints and Lipschitz constant $L > 0$. Then, there exists a rational neural network $R : [0,1] \to \mathbb{R}$ of size at most*

$$\mathcal{O}(m \log(\log(L/\epsilon)))$$

*such that $\max_{x \in [0,1]} |g(x) - R(x)| \leq \epsilon$.*

**Proof.** Let $0 \leq b_1 < \cdots < b_M \leq 1$ be the breakpoints of $g$. In a similar way to the proof of [14, Proposition 1], we first express $\rho$ as the following sum:

$$g(x) = c_0 \text{ReLU}(b_1 - x) + \sum_{j=1}^{m} c_j \text{ReLU}(x - b_j) + c_{m+1}, \tag{10}$$

for some constants $c_0, \ldots, c_{m+1} \in \mathbb{R}$. Therefore, $g$ can be exactly represented using a ReLU network with $m+1$ nodes and one layer, i.e.,

$$g(x) = \begin{pmatrix} c_0 & c_1 & \cdots & c_m \end{pmatrix} \begin{pmatrix} \text{ReLU}(-x + b_1) \\ \text{ReLU}(x - b_1) \\ \vdots \\ \text{ReLU}(x - b_m) \end{pmatrix} + c_{m+1}.$$

Since $g$ has a Lipschitz constant of $L$, we find that $|c_0| \leq L$ and $\sum_{j=1}^{m} |c_j| \leq L$. Using Lemma 1 we can approximate a ReLU function on $[-1,1]$ with tolerance $\epsilon/(2L)$ by a rational network $R_{\text{ReLU}}$ of size $\mathcal{O}(\log(\log(2L/\epsilon)))$. Now, we construct $R : [0,1] \to \mathbb{R}$ as a rational network obtained by replacing the ReLU functions in $g$ by $R_{\text{ReLU}}$. We have the following error estimate:

$$\max_{x \in [0,1]} |g(x) - R(x)| \leq |c_0| \|\text{ReLU} - R_{\text{ReLU}}\|_\infty + \sum_{j=1}^{m} |c_j| \|\text{ReLU} - R_{\text{ReLU}}\|_\infty \leq \frac{\epsilon}{2} + \frac{\epsilon}{2} \leq \epsilon.$$

The result follows as $R$ is of size $\mathcal{O}(m \log(\log(L/\epsilon)))$. ■

We remark that the size of the rational network required to approximate a piecewise linear function depends on $\epsilon$. In contrast, ReLU neural networks can represent piecewise linear functions exactly. In the next proposition, we show that a rational neural network can represent $x^n$, for some integer $n$, exactly.

**Proposition 6** *Let $n \geq 1$, $r_P \geq 2$, and $r_Q \geq 0$. There exists a rational network $R$, with rational activation functions of type $(r_P, r_Q)$, of size at most $5\lfloor \log_{r_P}(n) \rfloor^2 + 1$ such that $R(x) = x^n$ for all $x \in \mathbb{R}$.*

**Proof.** We start by expressing $n$ in base $r_P$, i.e.,

$$n = \sum_{\ell=0}^{\lfloor \log_{r_P}(n) \rfloor} c_\ell r_P^\ell, \qquad c_\ell \in \{0, 1, \ldots, r_P - 1\}.$$

This means we can represent $x^n$ as

$$x^n = \prod_{\ell=0}^{\lfloor \log_{r_P}(n) \rfloor} x^{c_\ell r_P^\ell}. \tag{11}$$

Note that $x^{c_\ell r_P^\ell}$ is just $x^{r_P}$ composed $\ell$ times as well as composed with $x^{c_\ell}$ so can be represented by a rational neural network with $\ell + 1$ layers, each with one node. Therefore, all the $x^{c_\ell r_P^\ell}$ terms can be represented in rational networks that in total have size

$$\sum_{\ell=0}^{\lfloor \log_{r_P}(n) \rfloor} (\ell + 1) = \frac{1}{2}(\lfloor \log_{r_P}(n) \rfloor)^2 + \frac{3}{2}\lfloor \log_{r_P}(n) \rfloor + 1.$$

The function $x^n$ can be formed by multiplying all the $x^{c_\ell r_P^\ell}$ terms together. Since $xy = (x^2 + y^2 - (x-y)^2)/2$, there is a rational network with one layer and three nodes that represents the multiplication operation. Therefore, multiplying all the terms together requires a rational network of size at most $3\lfloor \log_{r_P}(n) \rfloor$ (see Equation (11)). The result follows by noting that $x^2/2 + 9x/2 + 1 \leq 5x^2 + 1$ for $x \geq 1$. ■

Figure 1: Partition of unity: $\psi_0$ (red), $\psi_1$ (blue), and $\psi_2$ (green), for $N = 2$.

We can now prove Theorem 4 using the two previous propositions.

**Proof of Theorem 4.** The proof is based on the proof of [14, Theorem 1] and consists of replacing the piecewise linear functions and monomials arising in the local Taylor approximation of the function $f$ by rational networks using the previous approximation results.

Let $N \geq 1$ be an integer and consider a partition of unity of $(N + 1)^d$ functions $\phi_{\mathbf{m}}$ on the domain $[0, 1]^d$, i.e.,

$$\sum_{\mathbf{m} \in \{0,\ldots,N\}^d} \phi_{\mathbf{m}}(\mathbf{x}) = 1, \qquad \phi_{\mathbf{m}}(\mathbf{x}) = \prod_{k=1}^{d} \psi_{m_k}(x_k), \qquad \mathbf{x} = (x_1, \ldots, x_d),$$

where $\mathbf{m} = (m_1, \ldots, m_d)$, and $\psi_{m_k}$ is given by

$$\psi_{m_k}(x) = \begin{cases} 1, & \text{if } \left| x_k - \frac{m_k}{N} \right| < \frac{1}{3N}, \\ 0, & \text{if } \left| x_k - \frac{m_k}{N} \right| > \frac{2}{3N}, \\ 2 - 3N \left| x_k - \frac{m_k}{N} \right|, & \text{otherwise.} \end{cases}$$

Examples of the functions $\psi_{m_k}$ are shown in Figure 1 when $N = 2$. We now define a local Taylor approximation of $f$ by

$$f_N(\mathbf{x}) = \sum_{\mathbf{m} \in \{0, \ldots, N\}^d} \phi_{\mathbf{m}}(\mathbf{x}) P_{\mathbf{m}}(\mathbf{x}),$$

where $P_{\mathbf{m}}$ denotes the degree $n-1$ Taylor polynomial of $f$ at $\mathbf{x} = \mathbf{m}/N$. That is,

$$P_{\mathbf{m}}(\mathbf{x}) = \sum_{|\mathbf{n}| < n} \frac{D^{\mathbf{n}} f(\frac{\mathbf{m}}{N})}{\mathbf{n}!} \left( \mathbf{x} - \frac{\mathbf{m}}{N} \right)^{\mathbf{n}}, \tag{12}$$

where $|\mathbf{n}| = \sum_{k=1}^{d} n_k$, $\mathbf{n}! = \prod_{k=1}^{d} n_k!$, and $(\mathbf{x} - \mathbf{m}/N)^{\mathbf{n}} = \prod_{k=1}^{d} (x_k - m_k/N)^{n_k}$. Let $\mathbf{x} \in [0,1]^d$ and note that

$$\text{support}(\phi_{\mathbf{m}}) \subset \left\{ \mathbf{x} = (x_1, \ldots, x_d) : \left| x_k - \frac{m_k}{N} \right| < \frac{1}{N} \right\}, \qquad \mathbf{m} \in \{0, \ldots, N\}^d.$$

Hence, the approximation error between $f$ and its local Taylor approximation satisfies

$$|f(\mathbf{x}) - f_N(\mathbf{x})| = \left| \sum_{\mathbf{m} \in \{0, \ldots, N\}^d} \phi_{\mathbf{m}}(f(\mathbf{x}) - P_{\mathbf{m}}(\mathbf{x})) \right|$$

$$\leq \sum_{\mathbf{m}: \left| x_k - \frac{m_k}{N} \right| < \frac{1}{N}} |f(\mathbf{x}) - P_{\mathbf{m}}(\mathbf{x})|$$

$$\leq \frac{2^d d^n}{n!} \left( \frac{1}{N} \right)^n \max_{|\mathbf{n}|=n} \operatorname*{ess\,sup}_{\mathbf{x} \in [0,1]^d} |D^{\mathbf{n}} f(\mathbf{x})|$$

$$\leq \frac{2^d d^n}{n!} \left( \frac{1}{N} \right)^n.$$

We now select (see [14, Theorem 1] for a similar idea)

$$N = \left\lceil \left( \frac{n!}{2^d d^n} \frac{\epsilon}{2} \right)^{-1/n} \right\rceil,$$

so that

$$\max_{\mathbf{x} \in [0,1]^d} |f(\mathbf{x}) - f_N(\mathbf{x})| \leq \epsilon/2. \tag{13}$$

We now approximate the function $f_n$ by a rational network using Propositions 5 and 6. First, we write $f_N$ as

$$f_N(\mathbf{x}) = \sum_{\mathbf{m} \in \{0, \ldots, N\}^d} \sum_{|\mathbf{n}| < n} a_{\mathbf{m}, \mathbf{n}} \phi_{\mathbf{m}}(\mathbf{x}) \left( \mathbf{x} - \frac{\mathbf{m}}{N} \right)^{\mathbf{n}}, \tag{14}$$

where $|a_{\mathbf{m}, \mathbf{n}}| \leq 1$ and the monomials are uniformly bounded by 1 (see Equation (12)). Equation (14) consists of at most $d^n (N+1)^d$ terms of the form $\phi_{\mathbf{m}}(\mathbf{x})(\mathbf{x} - \mathbf{m}/N)^{\mathbf{n}}$. The monomial part $(\mathbf{x} - \mathbf{m}/N)^{\mathbf{n}}$ is representable by a rational network of size $\mathcal{O}(d \log(n)^2)$ using Proposition 6, including the fact that the multiplication is a rational network with one layer and three nodes. Let $0 < \delta < 1$ be a small number, for each $m_k \in \{0, \ldots, N\}$ the piecewise linear function $\psi_{m_k}$ has a Lipschitz constant of $L = 3N$. Therefore, it can be approximated with a tolerance $\delta$ by a rational network $\tilde{\psi}_{m_k}$ of size $\mathcal{O}(\log(\log(N/\delta)))$ (see Proposition 5). We can assume $\|\tilde{\psi}_{m_k}\|_{\infty} = 1$ by increasing the size of the network by a constant. This yields the following approximation error between a term

in Equation (14) and the rational network constructed using $\tilde{\psi}_{m_k}$:

$$\left| \phi_{\mathbf{m}}(\mathbf{x}) \left( \mathbf{x} - \frac{\mathbf{m}}{N} \right)^{\mathbf{n}} - \prod_{k=1}^{d} \tilde{\psi}_{m_k}(x_k) \left( \mathbf{x} - \frac{\mathbf{m}}{N} \right)^{\mathbf{n}} \right| \leq \left| \prod_{k=1}^{d} \psi_{m_k}(x_k) - \prod_{k=1}^{d} \tilde{\psi}_{m_k}(x_k) \right|$$

$$\leq \left| \psi_{m_1}(x_1) - \tilde{\psi}_{m_1}(x_1) \right| \left| \prod_{k=2}^{d} \psi_{m_k}(x_k) \right| + \left| \tilde{\psi}_{m_1}(x_1) \right| \left| \prod_{k=2}^{d} \psi_{m_k}(x_k) - \prod_{k=2}^{d} \tilde{\psi}_{m_k}(x_k) \right|$$

$$\leq \left| \psi_{m_1}(x_1) - \tilde{\psi}_{m_1}(x_1) \right| + \left| \prod_{k=2}^{d} \psi_{m_k}(x_k) - \prod_{k=2}^{d} \tilde{\psi}_{m_k}(x_k) \right|$$

$$\leq \delta + \left| \prod_{k=2}^{d} \psi_{m_k}(x_k) - \prod_{k=2}^{d} \tilde{\psi}_{m_k}(x_k) \right| \leq d\delta.$$

Here, the final inequality is derived by repeating the previous inequalities for $x_2, \ldots, x_d$. If we denote by $\tilde{f}_N$ the rational network approximation to $f_N$ constructed above, then, for all $\mathbf{x} \in [0,1]^d$, we have

$$|f_N(\mathbf{x}) - \tilde{f}_N(\mathbf{x})| \leq \sum_{\mathbf{m} \in \{0,\ldots,N\}^d} \sum_{|\mathbf{n}| < n} |a_{\mathbf{m},\mathbf{n}}| \left| \phi_{\mathbf{m}}(\mathbf{x}) \left( \mathbf{x} - \frac{\mathbf{m}}{N} \right)^{\mathbf{n}} - \prod_{k=1}^{d} \tilde{\psi}_{m_k}(x_k) \left( \mathbf{x} - \frac{\mathbf{m}}{N} \right)^{\mathbf{n}} \right|$$

$$\leq 2^d d^{n+1} \delta.$$

Therefore, we select $\delta = \epsilon/(2^{d+1} d^{n+1})$ so that $\max_{\mathbf{x} \in [0,1]^d} |f_N(\mathbf{x}) - \tilde{f}_N(\mathbf{x})| \leq \epsilon/2$. Then, by Equation (13), we have

$$\max_{\mathbf{x} \in [0,1]^d} \left| f(\mathbf{x}) - \tilde{f}_N(\mathbf{x}) \right| \leq \frac{\epsilon}{2} + \frac{\epsilon}{2} \leq \epsilon.$$

The statement of the theorem follows as the rational network $\tilde{f}_N$ has size at most

$$\mathcal{O}(d^n (N+1)^d \log(\log(N/\delta))) = \mathcal{O}(\epsilon^{-d/n} \log(\log(1/\epsilon^{1+1/n}))) = \mathcal{O}(\epsilon^{-d/n} \log(\log(1/\epsilon))).$$

∎

## C   Details of the approximation experiment

We use the TensorFlow implementation[1] of the deep hidden physics model framework to build and train the identifier network $\mathcal{N}$ that approximates a solution $u$ to the KdV equation. The true solution is computed on the domain $(x, t) \in [-20, 20] \times [0, 40]$ by Raissi [10] using the Chebfun package [4] with a spectral Fourier discretization of 512 and a time-step of $\Delta t = 10^{-4}$. Moreover, the solution is stored after every 2000 time steps, giving a testing data set of approximatively $10^5$ spatio-temporal points in $[-20, 20] \times [0, 40]$. We then constituted the training and validation sets (of $10^4$ points each) by randomly subsampling the solution at $2 \times 10^4$ points in $[-20, 20] \times [0, 40]$.

Table 1: Initialization coefficients of the rational activation functions.

| $a_0$ | $a_1$ | $a_2$ | $a_3$ | $b_0$ | $b_1$ | $b_2$ |
|-------|-------|-------|-------|-------|-------|-------|
| 1.1915 | 1.5957 | 0.5000 | 0.0218 | 2.3830 | 0.0000 | 1.0000 |

In a similar manner to [10], we use a fully connected identification network to approximate $u$ with 4 hidden layers with 50 nodes per layer. The network is trained using the L-BFGS optimization algorithm with 10,000 iterations. We compare three types of activation functions: ReLU, sinusoid, trainable rational functions of type $(3, 2)$, and trainable polynomials of degree 3. Furthermore, the rational activation functions are initialized to be the best approximation to the ReLU function (see Section 4), giving the initial coefficients reported in Table 1.

We represent the approximation errors between the different identification networks and the solution to the KdV equation in Figure 2.

Figure 2: Approximation errors of the neural networks with ReLU, sinusoid, and rational activation layers.

Finally, in Figure 3, we compare rational neural networks with different degree activation functions (each initialized to approximate the ReLU function using the MATLAB code `initial_rational_coeffs.m` available at [3]) and find that they all performed better than ReLU networks. While a type $(3, 2)$ rational offers a good trade-off between the number of parameters and quality of approximation according to the theoretical results presented in Section 3, the type of rational function might well depend on the application considered.

Figure 3: Validation loss of rational networks of types $(2, 2)$, $(3, 2)$, $(4, 3)$, and $(5, 4)$ with respect to the number of epochs.

## D    Details of the GAN experiment

We adapt the Keras example in [1] to train an Auxiliary Classifier GAN with rational activation functions on the MNIST. The hyper-parameters used for the GAN experiment are given in Table 2. Moreover, the GAN is trained on 20 epochs with a batch size of 100 by Adam's optimization algorithm [5] and the following parameters: $\alpha = 0.0002$ and $\beta_1 = 0.5$, as suggested by [9].

We report in Figure 4 samples of the 10 classes present in the MNIST dataset (right) and images generated at the 20th epoch by the GAN with ReLU/Leaky ReLU units (left) and rational activation functions (middle).

## Footnotes

[1]We adapt the code that is publicly available [11].

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

Figure 4: Forty images generated by a ReLU network and a rational network after 20 epochs, together with real images from the MNIST dataset.

Table 2: Hyper-parameters of the GAN experiment, BN denotes the presence of a Batch normalization layer. The Generator and Discriminator networks are trained with ReLU and rational activation functions, initialized with the coefficients reported in Table 1.

| Operation | Kernel | Strides | Features | BN | Dropout | Activation |
|---|---|---|---|---|---|---|
| Generator | | | | | | |
| Linear | N/A | N/A | 3456 | ✗ | 0.0 | ReLU / Rational |
| Transposed Convolution | $5 \times 5$ | $1 \times 1$ | 192 | ✓ | 0.0 | ReLU / Rational |
| Transposed Convolution | $5 \times 5$ | $2 \times 2$ | 96 | ✓ | 0.0 | ReLU / Rational |
| Transposed Convolution | $5 \times 5$ | $2 \times 2$ | 1 | ✗ | 0.0 | Tanh |
| Discriminator | | | | | | |
| Convolution | $3 \times 3$ | $2 \times 2$ | 32 | ✗ | 0.3 | Leaky ReLU / Rational |
| Convolution | $3 \times 3$ | $1 \times 1$ | 64 | ✗ | 0.3 | Leaky ReLU / Rational |
| Convolution | $3 \times 3$ | $2 \times 2$ | 128 | ✗ | 0.3 | Leaky ReLU / Rational |
| Convolution | $3 \times 3$ | $1 \times 1$ | 256 | ✗ | 0.3 | Leaky ReLU / Rational |
| Linear | N/A | N/A | 11 | ✗ | 0.0 | Soft-Sigmoid |

[5] Diederik P. Kingma and Jimmy Ba. Adam: A method for stochastic optimization. *arXiv preprint arXiv:1412.6980*, 2014.

[6] Harold W. Kuhn and Albert W. Tucker. Nonlinear Programming. In *Proc. Second Berkeley Symp. on Math. Statist. and Prob.*, pages 481–492. Univ. of Calif. Press, 1951.

[7] V. I. Lebedev. On a Zolotarev problem in the method of alternating directions. *USSR Comp. Math. Math+*, 17(2):58–76, 1977.

[8] Yuji Nakatsukasa and Roland W. Freund. Computing fundamental matrix decompositions accurately via the matrix sign function in two iterations: The power of Zolotarev's functions. *SIAM Rev.*, 58(3):461–493, 2016.

[9] Alec Radford, Luke Metz, and Soumith Chintala. Unsupervised Representation Learning with Deep Convolutional Generative Adversarial Networks. *arXiv preprint arXiv:1511.06434*, 2015.

[10] Maziar Raissi. Deep hidden physics models: Deep learning of nonlinear partial differential equations. *J. Mach. Learn. Res.*, 19(1):932–955, 2018.

[11] Maziar Raissi. GitHub repository. `https://github.com/maziarraissi/DeepHPMs/`, 2020.

[12] Matus Telgarsky. Neural networks and rational functions. In *Proceedings of the 34th International Conference on Machine Learning (ICML)*, volume 70, pages 3387–3393, 2017.

[13] N. S. Vyacheslavov. On the uniform approximation of $|x|$ by rational functions. *Sov. Math. Dokl.*, 16:100–104, 1975.

[14] Dmitry Yarotsky. Error bounds for approximations with deep ReLU networks. *Neural Netw.*, 94:103–114, 2017.