[Reviews · NeurIPS 2020]

Review 1

Summary and Contributions: Summary: this article considers approximation and initialization of neural networks with rational activation functions. It also provides some numerical evidence that such networks can give reasonable performance on some tasks.

Strengths: (1) I think approximation theory using neural networks is an interesting subject and the quantitative results for approximation with rational networks in Section 3 are certainly of interest to the approximation theory community. (2) The suggestion to initialize near a ReLU-approximant and then allowing the parameters of the rational function to be learned seems reasonable. This may have find some practical applications, though I think it is somewhat unlikely.

Weaknesses: (1) In general, I think articles that study the pure approximation power of neural networks, without regard for what can actually be learned from a random initialization can run the risk of being irrelevant to neural network community as a whole. That being said, as I mentioned above, I think there is enough here that, although this is indeed a weakness, it is not a fatal one. (2) The improvement for log(1/\eps) to log(log(1/\eps)) in Theorem 4 (for ReLUL compared with rational) doesn't strike me as particularly interesting given the \eps^{-d/n} in front is unchanged. (3) I am not entirely convinced by the numerical experiments. For example, in Figure 2, on the right (if I am understanding correctly is plotted the training loss). Lower training loss doesn't necessarily mean better test accuracy/loss, so it's hard to say that the rational networks are better from such a plot.

Correctness: Although I didn't check every detail, the claim appear to the correct.

Clarity: yes

Relation to Prior Work: yes

Reproducibility: Yes

Additional Feedback: POST REBUTTAL UPDATE: I take the authors' point about the near-optimality of \eps^{-d/n} from the DeVore et. al. paper of 1989. Note however, that in that work there is a continuity assumption on the parameter section and function reconstruction maps. Is you construction obviously continuous in the function? In any case, SGD-based parameter selection need not be continuous. Indeed, some work by Yarotsky shows that if we allow for discontinuous parameter selection then NNs can do way better than the \eps^{-d/n} error rate. I also understand the authors' point about the depth decreasing to \log(\log(1/\eps)). That indeed a good point. Although I would not say it is obviously of practical interest since no one says that you can learn this shallow representation in any numerically stable way. Finally, I am glad that the authors will put in a plot of validation accuracy. That is certainly helpful. My overall assessment remains positive (6/10).


Review 2

Summary and Contributions: This paper presents a neural network with a newly proposed activation function, avoiding the vanishing or improve the performance of deep learning models. This can be achieved by applying the rational function which has a good approximation capacity.

Strengths: 1. The solution for the raised problem is novel, i.e., rational function as activation. 2, strong theoretical study of the rational neural network 3, promising experimental results

Weaknesses: 1, Solution configuration is not well justified, e.g., the order of (3, 2) 2, Evaluation section only use GAN as a general deep learning model, could try more general models, and PDE is a specific problem. 3, Lack of comparison with popular polynomial approximation.

Correctness: yes; yes

Clarity: yes

Relation to Prior Work: yes

Reproducibility: Yes

Additional Feedback: This work proposes a new activation function to sever deep learning architecture, providing a theoretical study about its complexity. This paper is well-written and provides a high-level of readability to most readers of the data mining community. However, the article would be significantly enhanced if the issues related to their motivation, technical analysis, and experiments are addressed. Detailed comments are given in the following: 1) Motivation – This paper proposes rational activation function as an alternative to ReLU, potentially avoiding the issue of vanishing gradient problem * The problem raised in this paper, i.e., some existing activation functions (e.g., sigmoid, logistic) can only handle the smooth signal, is a significant problem in deep neural network optimization since their derivative are zero for large value. * The most popular activation, ReLU, has zero gradients for negative real values, and its performance lacks theoretical support. 2) Technical analysis – The paper proposes rational neural networks with a theoretical study about its approximation capacity and complexity. * The order of rational function is (3,2), but the justification is not provided. Low-degree can save time, but is there any better configuration and why choose such type? * There is another issue for (3,2) type since it can be reduced to a (2,2) plus a constant. That’s why rational function only has a larger denominator order; the reason why choosing a larger numerator order is not offered. * The result of Figure 1 shows the advantage of rational NN in approximating ReLU with little oscillation. However, the polynomial approximation is more popular techniques with less complexity and lower accuracy. The author may need to compare a rational neural network with polynomials. 3) Experiment – The proposed framework on PDE problem and GAN: * The result on PDE shows promising results, significantly faster than the baselines. GAN test also shows its superiority over ReLU. * However, rational neural networks are still a black box, and the authors may need to perform ablation or sensitivity test regarding the order. * Since the paper acclaims that it can help to solve the vanishing gradient issue, an additional experiment may be needed.


Review 3

Summary and Contributions: The paper studies neural networks equipped with rational activation functions. The authors begin with a good motivation on the importance of the activation functions in networks and list drawbacks of widely used activation functions, e.g., ReLU, tanh, sigmoid. Then rational activation functions of type (3, 2) is considered throughout the paper. Theoretical approximation theory illustrates rational functions can efficiently approximate ReLU functions, as well as nonparametric functions, e.g., Sobolev functions. Besides, empirical results on using rational networks for solving PDE and generating fake images of MNIST dataset demonstrates the improved performance of rational functions.

Strengths: Activation functions tie closely to the training and testing performance of neural networks. As the authors pointed out, smooth activation functions, e.g., sigmoid, tanh, can lead to gradient vanishing issue, while ReLU activation is only active for nonnegative inputs. The rational activation functions can potentially be a good alternative in some applications. From the theoretical results, the authors show that rational networks can more efficiently approximate smooth functions than ReLU networks. Even more importantly, the rational activation function used only has a degree smaller than 3. The empirical results also showcase the practicability of rational networks.

Weaknesses: The approximation theories of rational networks rely heavily on two works, Telgarsky and Yarotsky. In particular, the equivalence relation between ReLU activation and rational activation is based on the framework of Telgarsky, with an introduction of Zolotarev functions in replacement of the original Newman polynomials. This allows a tighter bound compared to Telgarsky. The universal approximation result is based on Yarotsky. The difference is Yarotsky uses ReLU network to approximate Taylor polynomials, while in this paper, rational networks are used. The technical contributions should be elaborated. For example, what is the core new steps in replacing Newman polynomials with Zolotarev functions? In addition, the improvement on the network size of using rational networks to approximate sobolev functions is marginal. In particular, the number of parameters in the network is reduced from \epsilon^{-d/n} \log 1/\epsilon to \epsilon^{-d/n} \log \log 1/\epsilon. The improvement of the dependence on \epsilon is highly likely only some constant multiplication. More importantly, the authors does not compare the constants hidden in the big O notation. The experiments also have some flaws in implementation, and are incomplete to demonstrate the strength of rational networks. 1. In solving PDE part, Line 244 reports the MSE of three different neural networks. This is presumably measured on the training set, rather on a testing set. Although this illustrates rational networks can potentially have better fitting ability (to confirm this, one should train three networks until stable, i.e., MSE does not visibly decrease), it is still a hasty conclusion that rational networks can perform better than ReLU and sinusoid networks (overfitting). 2. From the supplementary, the networks tested have the same architecture, and the only difference is the activation function. However, the total number of trainable parameters in these networks are not the same: rational activation functions bring more trainable weight parameters. One would suspect that the improved performance may due to the increased number of trainable parameters. 3. The experiment on GANs does not provide any quantitative results, therefore, the results does not add clarity to the performance of rational networks.

Correctness: The claims and method in the paper seems sound and correct, albeit not all the details are checked.

Clarity: The overall structure and flow of the paper is easy to follow. However, some mathematical claims are not rigorous. 1. Line 88: the big O notation hides a constant depending on the size and depth of the ReLU network --- the size of a network is a vague term and it is used across the theoretical claims. It is better to phrase it as the total number of trainable parameters or neurons in the network. 2. Zolotarev function is the key to show the claims in the paper, however, it is never defined and compared with Newman polynomials.

Relation to Prior Work: The contributions seem to be marginal, see the weakness section.

Reproducibility: Yes

Additional Feedback: -------------- Post Rebuttal --------------------- My main concerns of the paper are addressed in the response: 1) The authors provide experimental results on testing (validation), and demonstrate the good performance of using the rational activation compared with other common activation functions; 2) The relation with Telgarsky's work is highlighted in the response. I agree that the technique is different (this paper considers the compositional structure). Accordingly, I raise my rating to marginally above the threshold. One weakness of the paper still stands in my opinion. The improvement from \log 1/\epsilon to \log \log 1/\epsilon is rather marginal. Such an improvement does indicate the advantage of rational networks, and show rational networks can have smaller depth, though.


Review 4

Summary and Contributions: The paper investigates neural networks whose activation functions are (trainable) rational functions of their input. The paper shows theoretical results about the approximation power of these rational networks. In particular it shows that they improve on standard ReLU networks. The authors also perform experiments with rational activation functions of the type (3,2), i.e. polynomials of degree 3 at the numerator and polynomial of degree 2 at the denominator. The authors are interested in the future applicability of rational neural networks to partial differential equations (PDEs). Hence they study the KdV equation in their experiments.

Strengths: In the literature, most of the works that introduce “exotic” activation functions are motivated by empirical results, not supported by theoretical statements. In contrast with these works, the present paper proves mathematical results supporting the potentially improved approximation power of their fractional activation functions over ReLU.

Weaknesses: The authors note that the digits 1 generated by their rational network are all identical, while standard ReLU networks do not suffer from this problem. They point out that GANs are known to be hard to train and suggest that the rational GAN likely suffers from mode collapse. I appreciate the transparency of the authors, but I would have expected to see more compelling results on other tasks as well. I find the theoretical result of this work interesting, but from the experiments it is not clear yet that this result could have practical implications.

Correctness: The claims of this work are well grounded. The authors show in particular that: 1/ all ReLU networks can be epsilon-approximated by a rational network of size O(log(log(1/epsilon))) 2/ there exists a rational network that cannot be epsilon-approximated by a ReLU network of size less than O(log(1/epsilon)). These results suggest that rational networks could be more versatile than ReLU networks.

Clarity: The paper is well written overall.

Relation to Prior Work: The relation to prior work is adequately addressed. The authors explain that theoretical work by Telgarsky motivates the use of rational functions in deep learning. Molina et al. have experimented with rational activation functions (they defined the Padé Activation Unit), while Chen et al. propose to use high-degree rational activation functions in neural networks. In this work, the authors use low-degree rational functions as activation functions. Their composition in a deep network thus builds high-degree rational functions.

Reproducibility: Yes

Additional Feedback: The authors mention in several places of the paper “size” and “depth”. What is the definition of "size" here? (I would expect that size depends on depth) Is the architecture of the network assumed to be a regular MLP kind of neural net? (Can there be skip-layer connections for example?) ==== Post Rebuttal ==== Thanks to the authors for the nice rebuttal.

[Author Response · NeurIPS 2020]

**General comments.**   We thank the Reviewers for their detailed reviews and the feedback regarding the **theoretical study** of rational neural networks (NNs) and their **promising applications**. Reviewer 4 noted that the introduction of rational NNs is motivated by theory, while most exotic activation functions are only empirically supported. We address the referees' remarks on the theoretical results and numerical experiments here. The paper will be revised accordingly.

**Theoretical results.**   Reviewer 3 highlights the comparison between this paper and Telgarsky's work. We wish to emphasize that our key contribution is to employ a **composition of low-degree rationals** $r(x) = r_{\#\text{layers}} \cdots r_2(r_1(x))$, naturally realized by a NN, to approximate functions efficiently, while Telgarsky's work approximates a ReLU NN by a high-degree rational function not in the form of a NN. The Newman polynomials used by Telgarsky do not preserve (minimax) optimality under composition and result in an exponentially larger number of trainable parameters (see Fig. 1 of the paper for a comparison). Finally, we present **optimal lower and upper approximation bounds**. The notion of size and Zolotarev functions will be explicitly defined in the revised version, as requested by Reviewers 3 and 4.

In response to Reviewer 2, concerning the choice of the degree $(3, 2)$, we emphasize that **this degree appears naturally in the technical analysis** due to the composition property of the Zolotarev functions (see Section 3.1): the degree of the overall rational function $r$ is a whopping $3^{\#\text{layers}}$, while the **number of trainable parameters only grows linearly**. A superdiagonal degree $(3, 2)$ allows $r$ to behave like a **nonconstant function at** $\pm\infty$, unlike a diagonal degree e.g. $(2, 2)$. The theory ensures that low-degree rational activation functions minimize the number of trainable parameters given a fixed overall degree. The choice is also motivated empirically, and we do not claim that the degree $(3, 2)$ is the best choice for all situations as the best configuration may well depend on the application; see below Fig. 1 (right).

Reviewers 1 and 3 discussed the apparently marginal difference between $\log(1/\epsilon)$ and $\log\log(1/\epsilon)$ in Thm. 4. To clarify, the bound for rational NNs is **close to optimal**, given by $\epsilon^{-d/n}$ (DeVore et al., 1989). Most importantly, a rational NN can achieve this approximation power with a depth of only $\log\log(1/\epsilon)$, which is **exponentially smaller** than the $\log(1/\epsilon)$ layers needed by a ReLU network to approximate a smooth function to within $\epsilon$. This improvement is obtained from the composition of low-degree rational functions and is not hidden in multiplicative constants, which do not depend on $\epsilon$. This improved approximation power has **practical consequences** for larger NNs given that a deep NN is computationally expensive to train due to expensive gradient evaluation and slower convergence. The constants inside the $\mathcal{O}$ notations are computed whenever possible (see Lem. 1 and Cor. 3 of the Supplementary Material) but the main theorems of the paper treat a general setting, for which few (if any) papers in the literature give explicit constants.

**Experimental results.**   We thank the Reviewers for their comments on the experiment in Section 3.1. While Fig. 2 (right) of the paper showed that rational NNs are easier to train than ReLU and Sinusoid NNs, it did not show the accuracy loss. We have performed **new experiments with a validation set** independent of the training set and display the validation loss throughout training (see Fig. 1 (left)). We find that **rational NNs outperform the other NNs** during the training phase and on the testing set. Reviewer 3 was concerned that this result was due to the difference in number of trainable parameters but the ReLU NN had $8000$ and the rational had $7 \times \#\text{layers} + 8000 = 8035$. **Rational NN are never much bigger in terms of trainable parameters** than ReLU NNs since the increase is only linear with respect to the number of layers. Reviewer 2 suggested a **comparison with polynomial approximation**, which we have performed (see Fig. 1 (left)). Here, we train a NN with degree 3 polynomial activation functions. We observe that this NN is **harder to train than rational NNs** as shown by the non-smooth validation loss. Polynomials perform poorly on non-smooth functions such as ReLU, with an algebraic convergence of $\mathcal{O}(1/\text{degree})$ (Trefethen, 2013) rather than the (root-)exponential convergence with rationals. Finally, following Reviewer 2's suggestion **we compare rational NNs with different degree activation functions** and find that they **all perform better than ReLU NNs** (see Fig. 1 (right)).

Figure 1: Left: Validation loss of a ReLU (blue), sine (green), polynomial (purple) of degree 3, and rational (red) NNs of type $(3, 2)$ with respect to the number of optimization steps. Right: Comparison between the validation losses of rational neural networks of types $(2, 2)$, $(3, 2)$, $(4, 3)$, and $(5, 4)$.

[Meta-Review · NeurIPS 2020]

The paper studies rational DNNs --- deep neural networks where rational functions (of small degrees) are used as non-linearities. The paper provides many interesting theoretical results on the approximation properties of the rational DNNs (specifically, in comparison to ReLU DNNs). The paper also provides two experiments (learning the solution of the 2-dimensional PDE and applications in generative adversarial networks), which are meant to demonstrate that rational activations have advantages compared to other popular activations (ReLu, sine, tanh, polynomial, etc) when used in actual DNN training. The theory presented in the paper establishes that: (1) Consider two problems: (i) Approximating (in the uniform norm) a function implemented with the rational DNNs using ReLU DNNs; and (ii) approximating a function implemented with the ReLU DNNs using rational DNNs. Theorem 3 shows that (ii) is much easier than (i): (ii) can be solved to eps-precision with log(log(1 / eps)) many parameters, whereas (i) requires at least log(1 / eps ) parameters (exponentially more). (2) Theorem 4 shows that any function in a specific Sobolev space can be eps-approximated with the rational DNNs using log(log(1/eps)) eps^{-d/n} parameters, where the inputs are d-dimensional and n controls the smoothness of the Sobolev space. Meanwhile, it has been previously known that in the case of ReLU networks same problem can be solved with log(1/eps) eps^{-d/n} parameters. These results are interesting for the approximation community. Their proofs are novel and use an interesting observation that the composition of rational functions of small degrees can lead to the rational function of a very large (exponential in the number of layers) degree while having relatively small (linear in the number of layers) number of parameters. My major concern (shared by most of the reviewers) is that these approximation results are not very relevant to the actual DNN training. The fact that there exists a parameter configuration that approximates a given function with the rational DNN does not mean that the same parameter configuration can be efficiently learned using SGD (or any other practical method). The only results supporting a relevance of the presented approximation theory are two experiments. The first one is essentially an MSE regression with two-dimensional inputs (Section 4.1, Figure 2) and 10k points in the training set. In the second experiments the authors train DCGAN-style generator-discriminator pair on MNIST with ReLU/LeakyReLU activations (used in the original DCGAN paper) replaced with the proposed rational activations. The results of the first experiment are convincing: indeed, the test MSE of the rational DNN decreases considerably faster than for any other considered activation function. Unfortunately, I am not sure if the success in the 2-dimensional MSE regression with 10k points in the training set (which is *a lot*) will transfer to more practical settings. The authors could at least train the rational DNNs on MNIST and compare this to the vanilla ReLU DNN training. I wonder if the authors tried this natural experiment. Results of the second GAN experiment are difficult to interpret: the authors did not provide any quantitative metrics (eg. Frechet Inception Distance, etc) and instead base their evaluation on the visual inspection. In the field of unsupervised generative modeling it has been established a long time ago (at least since [1]) that any results on GANs necessarily need to be supplemented with at least some quantitative evaluation metrics. Otherwise, the visual evaluation can lead to any desired conclusions. In summary, if the results on the pure approximation theory of rational DNNs are interesting enough for the NeurIPS community, I think the paper provides novel/strong/useful contributions. Otherwise, the relevance of the results to the DNN training is not clear at this point. [1] Are GANs Created Equal? A Large-Scale Study, 2017.